# Incidence and Predictors of Cardiac Implantable Electronic Devices Malfunction with Radiotherapy Treatment

**DOI:** 10.3390/jcm11216329

**Published:** 2022-10-27

**Authors:** Meor Azraai, Daisuke Miura, Yuan-Hong Lin, Thalys Sampaio Rodrigues, Voltaire Nadurata

**Affiliations:** 1Department of Cardiology, Bendigo Health, Bendigo, VIC 3550, Australia; 2Faculty of Medicine, Nursing and Health Sciences, School of Rural Health, Monash University, Melbourne, VIC 3550, Australia; 3Department of Radiation Oncology, Peter McCallum, Bendigo Health, Bendigo, VIC 3550, Australia; 4Faculty of Medicine, Dentistry and Health Sciences, University of Melbourne, Melbourne, VIC 3010, Australia

**Keywords:** cardiac implantable electronic devices, pacemaker, implantable cardioverter defibrillator, cardiac resynchronization therapy, arrythmia, radiotherapy, cancer

## Abstract

Aims: To investigate the incidence of cardiac implantable electronic devices (CIED) malfunction with radiotherapy (RT) treatment and assess predictors of CIED malfunction. Methods: A 6-year retrospective analysis of patients who underwent RT with CIED identified through the radiation oncology electronic database. Clinical, RT (cumulative dose, dose per fraction, beam energy, beam energy dose, and anatomical area of RT) and CIED details (type, manufacturer, and device malfunction) were collected from electronic medical records. Results: We identified 441 patients with CIED who underwent RT. CIED encountered a permanent pacemaker (PPM) (78%), cardiac resynchronization therapy—pacing (CRT-P) (2%), an implantable cardioverter defibrillator (ICD) (10%), and a CRT-defibrillator (CRT-D) (10%). The mean cumulative dose of RT was 36 gray (Gy) (IQR 1.8–80 Gy) and the most common beam energy used was photon ≥10 megavolt (MV) with a median dose of 7 MV (IQR 5–18 MV). We further identified 17 patients who had CIED malfunction with RT. This group had a higher cumulative RT dose of 42.5 Gy (20–77 Gy) and a photon dose of 14 MV (12–18 MV). None of the malfunctions resulted in clinical symptoms. Using logistic regression, the predictors of CIED malfunction were photon beam energy use ≥10 MV (OR 5.73; 95% CI, 1.58–10.76), anatomical location of RT above the diaphragm (OR 5.2, 95% CI, 1.82–15.2), and having a CIED from the ICD group (OR 4.6, 95% CI, 0.75–10.2). Conclusion: Clinicians should be aware of predictors of CIED malfunction with RT to ensure the safety of patients.

## 1. Introduction

Globally, patients with cardiac implantable electronic devices (CIED), which include permanent pacemakers (PPM), implantable cardioverter defibrillators (ICD) and cardiac resynchronization therapy (CRT), are a common encounter in current clinical practice [1] Indications for CIED implantation are growing and range from treatment of symptomatic bradycardia, therapy for ventricular arrhythmias, and utilization in the management of heart failure with reduced ejection fraction [2].

Modern CIEDs use the latest complementary metal-oxide semiconductor (CMOS) integrated circuits. This technology allows the miniaturization of the device while prolonging the battery life of the device as CMOS circuits have low energy consumption [3]. However, these circuits are more susceptible to interference from external energy sources, such as therapeutic doses of radiotherapy (RT) used in cancer treatment [4].

The global cancer burden is on the rise as a result of the world’s population growth and ageing [5]. Ionizing radiation from external beam RT and neutron-producing therapy (especially photons) used in the treatment of cancers can interfere with CIED function, with potential serious harm to patients [6,7]. An extensive literature review performed by our group recognized that RT dose to CIED > 5 grays (Gy) and photon beam energy of ≥ 10 megavolts (MV) carries the highest risk of CIED malfunction [8].

Despite the recognized effect of RT on CIED, CIED manufacturer guidelines differ from one brand to another, and there is a paucity of large-scale clinical studies investigating this patient cohort. Our study aims to add useful clinical data to this growing patient group by investigating the incidence of CIED malfunction, factors that are associated with an increased risk of CIED malfunction, and resulting clinical consequences in patients undergoing RT.

## 2. Methods

### 2.1. Study Design and Setting

Ethical approval was provided by the Human Research Ethics Committee at Bendigo Health, Victoria. We conducted a single-center retrospective study of patients with CIED who received RT as treatment for their cancer in the radiation oncology department located in Bendigo Hospital from December 2016 to December 2021. Bendigo Hospital is the biggest major regional hospital in Victoria, Australia, and covers a large catchment area that spans up to 200 km northwest of Bendigo, Victoria. Our radiation oncology unit is supported by a well-established cardiac electrophysiology (EP) unit in the cardiology department. All patients with CIED undergoing RT are notified to the EP unit. A CIED technician will be present for most of the RT course, according to their risk stratification, for the notified patients.

### 2.2. Participants

All adult patients (age ≥ 18 years) undergoing radiotherapy who have an implanted cardiac device (PPM, ICD, or CRT) were eligible.

We used the radiation oncology electronic database to identify patients who received RT from 1 December 2016 to 1 December 2021. We then identified patients with CIED as having an ICD-10 (The International Classification Diagnosis of Disease 10th Version) discharge diagnosis of Z95.0 and Z95.810. We then divided patients with CIED who underwent RT into those with CIED malfunction and those without CIED malfunction. Patients with inadequate documentation, unable to complete the RT course, followed up with another hospital, and lost to outpatient follow up were excluded.

### 2.3. CIED Data

Data on CIEDs were collected from Cardiobase^®^, an encrypted software that contains reports of all cardiac investigations that were performed at Bendigo Hospital. Cardiobase^®^ includes clinical information on device class [PPM, cardiac resynchronization therapy—pacemaker (CRT-P), ICD, or CRT-defibrillator (CRT-D)], device type (single chamber PPM, dual chamber PPM, CRT-P, CRT-D, or ICD), device manufacturer (Medtronic, St Jude, Biotronik, St Jude), indication for CIED implantation and follow-up device interrogation. Device dependency was defined as patients requiring a functioning CIED to avoid cardiovascular collapse. Patients who were considered as being CIED dependent for this study were those with an indication for CIED insertion for sinus bradycardia <30 beats per minute, sinus pauses, type II Mobitz block, complete heart block, primary/secondary prevention of ventricular arrhythmia, and use of CRT [9].

### 2.4. Radiation Oncology Data

Details on clinical data, cancer, RT and beam energy treatments were collected from the Bendigo Hospital electronic medical records. Each patient with CIED undergoing RT had a pre-RT assessment form (Figure 1) completed, which contains most information.

RT treatment details include anatomical site of treatment, total RT dose, number of RT fractions, type of beam energy, and beam energy dose. “Actions to be taken” section includes cardiac monitoring required prior or during treatment, deactivation/reprogramming of device during treatment, relocation of device or no actions required.

As we aim to ensure the privacy of patients and healthcare professionals involved, the names, date of birth, identification number, planner name, and medical physicist involved were not collected from this form.

### 2.5. CIED Malfunction

We examined CIED interrogation performed throughout RT treatment (prior, during, and at least 3 months after last RT) through Cardiobase^®^, to evaluate the incidence of CIED malfunction and if malfunction resulted in a clinically significant event. CIED malfunctions were categorized as follow:(1)Electrical reset to backup mode or other minor software error.(2)Electrical reset or other software error requiring reprogramming of CIED by the manufacturer.(3)Unexpected decrease in battery life capacity.(4)Loss of telemetry.(5)Change in one or several lead parameters eventually resulting in follow-up visits or lead replacement.(6)Noise oversense without symptomatic pacing inhibition, antitachycardia pacing (ATP), or shock therapy.(7)Oversense with symptomatic pacing inhibition, ATP, or shock therapy.

Clinically significant events that were observed were dizziness, syncope, dyspnoea, chest pain, hypotension, bradycardia, or tachycardia.

### 2.6. Statistical Analysis

Statistical analysis was performed using IBM SPSS, version 26 (IBM Corp, Armonk, NY, USA). For normally distributed continuous data, we reported the mean and standard deviation. For data with a skewed distribution, we reported the median and interquartile range. Categorical variables will be presented as frequency (percentage). Differences in characteristics between groups (patients with device malfunction vs. without device malfunction) were assessed using independent sample Student’s *t*-test or Mann–Whitney U test for continuous variables and chi-squared or Fisher’s exact tests for categorical variables, as appropriate.

The odds ratios (ORs) with 95% confidence intervals (Cis) of CIED malfunctions were computed using logistic regression. Independent variables in the model were a type of device (ICD/CRT-D vs. PPM/CRT-P), anatomical region irradiated (above vs. below diaphragm), cumulative radiation dose (in 10 Gy increments), fraction dose (in 1 Gy increments), and beam energy (≥10 MV vs. <10 MV). The cut-off value of 10 MV for beam energy was chosen based on clinical guidelines [10,11] that states a beam energy of ≥10 MV increases the risk of CIED malfunction. ORs were adjusted for beam energy. Anatomical regions irradiated above the diaphragm are RT to the head and neck, thorax, and upper limbs. The anatomical region below the diaphragm consists of RT to the abdomen and pelvis, spine, and lower limbs.

As some patients received more than one RT course, the RT courses were not completely independent. To accommodate for this dependence, the method of generalized estimating equations was used in a generalized linear model.

## 3. Results

### 3.1. Descriptive Characteristics

There were 441 consecutive patients with CIED who had RT in Bendigo Hospital from December 2016 to December 2021. The search strategy, along with the number of eligible and excluded patients is shown in Figure 2. We then identified 17 patients who had CIED malfunction with RT therapy and 424 patients who did not have CIED malfunction with RT therapy.

The baseline characteristics, CIED data, and RT details of all patients with CIED who underwent RT are summarized in Table 1. The median age for patients undergoing RT was 82 ± 14 years old and were predominantly male, 327 (74%) patients. The most common CIED type was dual chamber PPM, with 300 (69%) devices, with the majority of the CIEDs manufactured by Medtronic (45%). Nearly half, 216 (49%), of the patients were device dependent, but there were only 84 (19%) patients who had safety measures performed. Application of a magnet was most commonly used in patients with ICD, with 49 (58%) patients having this safety measure performed. No surgical relocation of CIED was required in our study.

The most common anatomical location for RT was the head and neck, 137 (31%) patients, which was mostly used for treatment of skin cancers, in 96 patients. The cumulative dose varied widely, with a median of 36 Gy (IQR 1.8–80 Gy), which is reflected in the dose per fraction as well, with a median of 3.75 (1–26 Gy). The dose to the device was consistent, at 0.28 Gy (IQR 0.1–3.3 Gy). The most common type of beam energy used was photon ≥10 MV, with usage in 185 (44%) patients out of 423 patients being treated with beam energy.

### 3.2. Patients with CIED Malfunction vs. without CIED Malfunction with RT

The CIED type with the most malfunctions was the defibrillator (ICD and CRT–D), which amounted to 11 (65%) of the CIED malfunctions. Medtronic CIEDs were associated with the most malfunctions [9 (53%) devices], but this was expected as Medtronic devices were the most commonly used CIEDs. It is important to note that in all patients who had CIED malfunction, no safety measures were utilized.

CIED malfunction more frequently developed in patients who had RT to the thorax, with 12 (71%) patients affected. The cumulative dose and dose per fraction were higher in patients who had CIED malfunction, at 42.5 Gy (20–77 Gy) vs. 36 MV (1.8–80 Gy) and 12 Gy (10–26 Gy) vs. 3.6 Gy (1–24 Gy), respectively. The number of fractions is lower, 6 (2–23) vs. 12.9 (1–40) fractions, in those with CIED malfunction, which directly correlates with a higher dose per fraction. Dose to device is fairly similar in those with and without a CIED malfunction.

The most obvious difference between the two groups was usage of photon beam energy ≥ 10 MV, which was associated with 16 (94%) CIED malfunctions with a higher median beam energy dose, 14 MV (12–18 MV) vs. 6.75 MV (5–15 MV), in those without CIED malfunctions. Electron beam therapy utilized in 93 (22%) patients was not associated with any CIED malfunctions.

### 3.3. CIED Malfunction

All CIED malfunctions did not result in any clinically significant events. The types of CIED malfunctions that occurred with RT are listed in Table 2. Soft reset occurred in 5 (29%) CIEDs, affecting only the ICD group of CIEDs. Noise oversense with no symptoms predominantly occurred in the ICD group as well, where five ICDs out of six CIEDs experienced this malfunction.

Devices in the PPM group were most commonly affected by an unexpected decline in battery life, affecting three out of four devices, and a change in lead parameters, where the two affected CIEDs were PPMs. In our study, a reduction in pacing output occurred in one PPMs exposed to the highest dose of cumulative RT dose of 77 Gy with a dose per fraction of 26 Gy for a palliative RT procedure. In this case, device malfunction occurred without any use of beam energy and no fur.

Interestingly, noise oversense was associated with a higher photon beam energy dose, 15.6 MV (14–17 MV), and battery depletion was associated with a higher RT cumulative dose, 48.75 Gy (47–50 Gy). No noise oversense malfunction has resulted in syncope, hypotension, ATP treatment, inappropriate shock, or loss of pacing. An unexpected decline in battery life has led to the replacement of a generator in two out of four devices. Details of CIED malfunctions are listed in Appendix A.

### 3.4. Predictors of CIED Malfunction

Crude logistic regression analysis showed that CIED malfunctions were statistically associated with the usage of photon beam energy of ≥10 MV (OR 6.73; 95% CI, 1.58–10.76, *p* = 0.007), anatomical location of RT above the diaphragm (OR 5.2, 95% CI, 1.82–15.2, *p* = 0.014) and exposure to CIED from the ICD group (OR 4.6, 95% CI, 0.75–10.2, *p* = 0.021) (Table 3).

Interestingly, there was no significant correlation detected between CIED malfunction and cumulative RT dose (OR 1.20, 95% CI, 0.95–1.52, *p* = 0.31) or fraction dose (OR 1.0, 95% CI, 0.79–4.7, *p* = 0.43).

We did not include safety measures as a predictor of CIED malfunction, even though all patients with CIED malfunction did not have any safety measures applied. Application of magnet and reprogramming of the device does not prevent any device malfunction in the system, but only prevents device malfunction from manifesting clinically.

## 4. Discussion

To our knowledge, few studies have measured the incidence of CIED malfunctions with RT in clinical practice [12,13]. Our study has shown an increased risk of CIED malfunction in patients who have been exposed to photon beam energy of ≥10 MV, RT treatment to the region above the diaphragm, and with a CIED from the ICD group. These findings are consistent with other studies which showed that higher beam energy and RT treatment above the diaphragm (closer to the CIED) were associated with device malfunction [12,13]. Most of these findings are similar to recommendations made in clinical guidelines [10,11] and review articles [8,14] regarding lowering beam energy use and increasing the distance of the RT field from CIED.

None of the patients with CIED malfunction suffered from symptomatic malfunction, such as inappropriate shock or syncope, and no surgical relocation was required prior to RT, which is comparable to other studies [12]. Literature reviews performed on the effects of RT on CIED found clinically significant events from the CIED malfunction that occurred in CIEDs implanted prior to 2004 [8,11,12]. The decrease in clinically significant CIED malfunctions coincides with the release of recent clinical guidelines in patients with CIED undergoing RT [15], where a lower RT dose and shielding were recommended.

Photon beam energy use of ≥10 MV was the strongest predictor of CIED malfunction. Photon beam energy causes neutron-induced upsets in memory circuits by changing stored values through a single event upset (SEU) in the CMOS circuit. This causes transformation in the microprocessor circuit (resets) without any physical damage to the device [16]. The risk of SEU error is stochastic in nature [17], which is reproduced in our study with resets occurring in a wide range of photon beam energy (10–17 MV) with varying locations of RT from the head, neck (n = 2), thorax (n = 3), abdomen, and pelvis (*n* = 1). Electron therapy does not produce any malfunctions in other studies [8,12], including ours, as electron therapy produces few neutrons, which reduces the risk of SEU [18].

Higher cumulative doses of RT were not significantly associated with the CIED malfunction, which was reflected in multiple studies [12,13], including our study as well. Clinical guidelines recommend keeping RT dose to device < 5 Gy [10,11,15] due to evidence from in vivo and in vitro studies of CIED malfunction with RT in the past, which used very high RT cumulative doses of ≥50 Gy [19]. In light of the growing evidence and the reduction in average RT dose used in clinical practice, this recommendation should be revised in the future when some CIED manufacturers reduce their recommended RT dose to devices to be kept at <2 Gy (Table 4).

In our study, CIED with asymptomatic noise oversense was associated with a higher photon beam energy dose (14–18 MV) with a modest RT dose range (30–45 Gy). As beam energy is not sensed as myocardial potential but RT signals can be interpreted as a cardiac signal [20]. This malfunction was unlikely due to beam energy. We hypothesize that it is due to the relatively high RT dose used and the unmeasured electromagnetic noise associated with the linear accelerator (LINAC) of the radiotherapy beam generator [19]. Electromagnetic interference around modern LINACs is minimal and there is little concern with transient effects for CIEDs [19].

CIED with a change in lead parameters and an unexpected decline in battery life was associated with higher RT cumulative doses (47–77 Gy). A high cumulative RT dose can lead to CIED circuit degradation in proportion to the accumulated dose, which can result in decreased output amplitude and increased current drain [21]. These malfunctions are also permanent to the CIED and has resulted in a generator change in our study.

There was a CIED malfunction with reduced pacing lead output in a patient undergoing palliative RT for a hemorrhaging bronchial carcinoma where the cumulative dose was 77 Gy without any photon therapy. In a situation where a very high RT dose is utilized, there is a higher risk of CIED malfunction evident in cases where the RT delivered dose is ≥50 Gy [19]. Fortunately, high doses of RT are only used in palliative procedures where immediate symptom relief outweighs the possible damage to CIED. Much lower doses are commonly used in common thoracic cancers, such as lung or breast cancer [22].

There was a statistically significant finding that CIED malfunctions occurred more often in the ICD group, where 11 (65%) ICDs had malfunctions, compared to 4 devices in the PPM group. This finding is comparable to another study where there were malfunctions in 13 ICDs out of 20 CIEDs exposed to a photon dose of ≥15 MV and a RT cumulative dose of 80 Gy [23]. A 2003 recommendation by Guidant (part of Boston Scientific) suggested that ICDs may be up to 10 times more sensitive to radiation damage than PPM since operation instructions are stored in random access memory (RAM) [24]. A more complex electronic circuit with a greater number of materials (i.e., ^10^B or ^6^Li) producing ionizing particles interacting with neutrons can be found in ICD than in PPM. However, these statements are highly speculative and there needs to be more studies focusing on different devices that are more susceptible to RT and beam energy.

The CIED manufactured by Medtronic had the most malfunctions, affecting 9 out of 17 devices. Medtronic CIED was also the most common cardiac device to be implanted in our study, which might explain why malfunctions were more frequent with this brand. Manufacturer’s recommendations differ from one another (Table 4) and need to be streamlined or referred to a gold-standard clinical guideline [10,11] to avoid discrepancy in RT dose delivery and a likely imbalance of CIED malfunctions between manufacturers.

**Table 4 jcm-11-06329-t004:** Summary of recommendations from the major PM/ICD manufacturers regarding safe radiotherapy in CIED patients. CIED, cardiac implantable electronic device; Gy, gray; ICD, implantable cardioverter defibrillator; MV, megavolt; PM, pacemaker; RT, radiotherapy.

Summary of Recommendations from the Major CIED Manufacturers Regarding Safe Radiotherapy in CIED Patients				
Recommendations	Medtronic [25]	St Jude [26]	Boston Scientific [27]	Biotronik [28]
Device Checks				
Before RT course	Not stated	Not stated	Specific to each patient	Yes
During RT course	Yes (if exceed recommended safe dose)	Yes (a detailed evaluation once or twice during the RT course in PM-dependent patients)	Specific to each patient	Not stated
After RT course	Yes	Yes	Yes, including subsequent close monitoring of the device function	Yes, including a supplementary follow-up shortly after RT
Maximal PM dose	5 Gy	No safe dose	No safe dose (2 Gy used as a reference)	2 Gy
Maximal ICD dose	1–5 Gy depending on model	No safe dose	No safe dose (2 Gy used as a reference)	2 Gy
Maximal beam energy	≤10 MV	Not stated	Not stated	≤10 MV
Inactivation of antitachycardia therapies	Yes	Yes	Yes	Yes
Lead shielding of device	No (ineffective against neutrons)	Not stated (reduction in device dose is recommended)	All available shielding options	Yes
Heart rhythm monitoring during RT	Not stated	Yes	As determined most appropriate by the physician team	Yes

## 5. Strengths and Limitations

To our knowledge, this is the largest study on RT in CIED patients published to date. This study collected data that was not collected in other studies, such as CIED manufacturer, device dependency, and safety measures.

There was high mortality among the patients. Approximately one third of the devices were never evaluated after RT. Additionally, being outside the scope of the study, the causes of death were not analyzed in this subgroup. Thus, we may have underestimated both the occurrence and degree of severity of RT-induced device malfunctions. However, as beam energy and the proportion of ICDs were lower in RT courses without evaluation, we find it unlikely that we underestimated the occurrence to a major extent. In this study, we cannot rule out transient asymptomatic effects of radiation on the devices that were not detectable at a subsequent CIED evaluation.

## 6. Conclusions

This study has shown that CIED malfunctions with RT, which range from software resets, unexpected battery decline, lead damage, and noise oversense, did not lead to a clinically significant event. The predictors for CIED malfunctions are usage of photon beam energy ≥10 MV, the anatomical location of RT above the diaphragm, and the presence of ICD/CRT-D. However, cumulative RT dose has no significant correlation with device malfunction, despite the traditional assumption that RT causes malfunctions in CIEDs during treatment.

Based on the increasing number of patients with CIEDs and the growing number of cancers requiring RT treatment, clinicians need to be aware of such complications. Based on the growing evidence, specific consideration for beam energy should be implemented as a part of the initial assessment for patients with CIEDs who are undergoing RT.

## 7. Clinical Perspective

Competency in medical knowledge

Our study demonstrates that photon beam energy ≥10 MV, RT above the diaphragm and ICDs are significantly associated with CIED malfunction. Traditionally, the cumulative RT dose was thought to be the predominant cause of CIED malfunction. The results highlight the need to observe beam energy dose and to institute replanning of radiotherapy to reduce the risk of CIED malfunction. The CIED malfunctions that occurred in our study were minor malfunctions and did not result in any clinical symptoms.
Translational outlook

Future studies are needed to: (1) create a clinical pathway with specific consideration for beam energy during assessment for patients undergoing RT; and (2) determine the best approach (redirection of RT beam) if the tumor is located above the diaphragm or implementation of safety measures. In addition, more in vivo research on the reasons for the increased susceptibility of ICDs to RT or beam energy is needed to understand the mechanism to improve ICDs’ resistance to RT or beam energy.

## Figures and Tables

**Figure 1 jcm-11-06329-f001:**
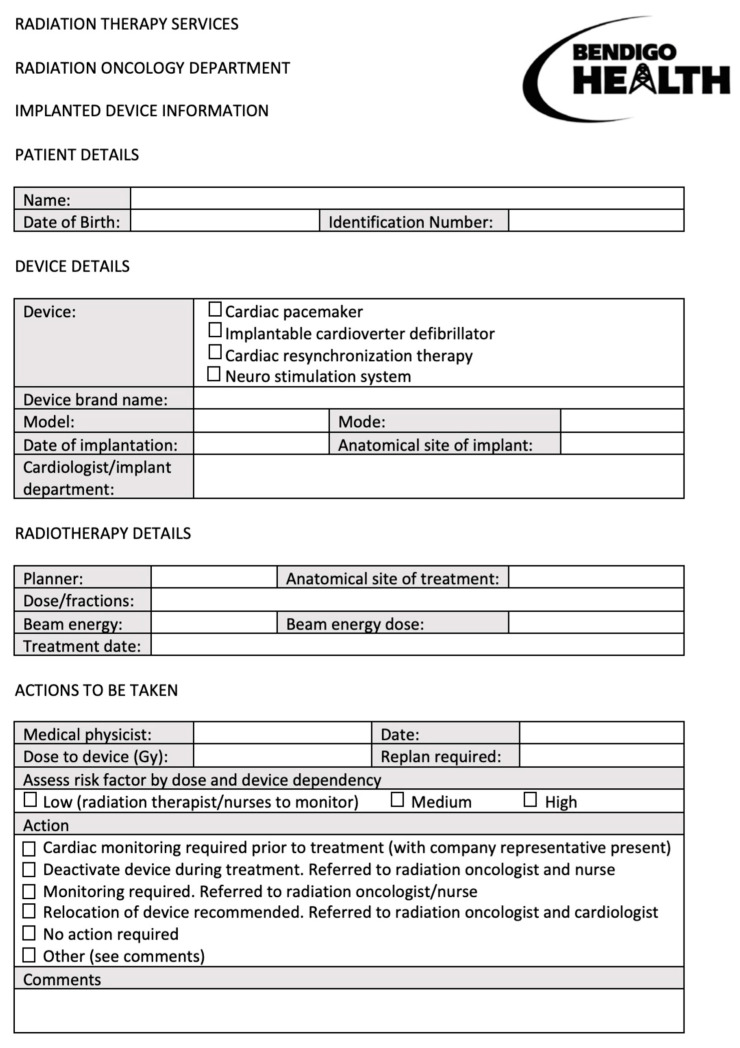
Pre-RT assessment form used in Bendigo Hospital. This form includes important information, such as details of CIED (device type, brand, mode, and site of implant) and RT (anatomical site of treatment, dose/fractions, dose to device, beam energy, and beam energy dose). This form concludes with a recommendation from the medical physicist, after evaluation of the CIED malfunction risk from the RT data.

**Figure 2 jcm-11-06329-f002:**
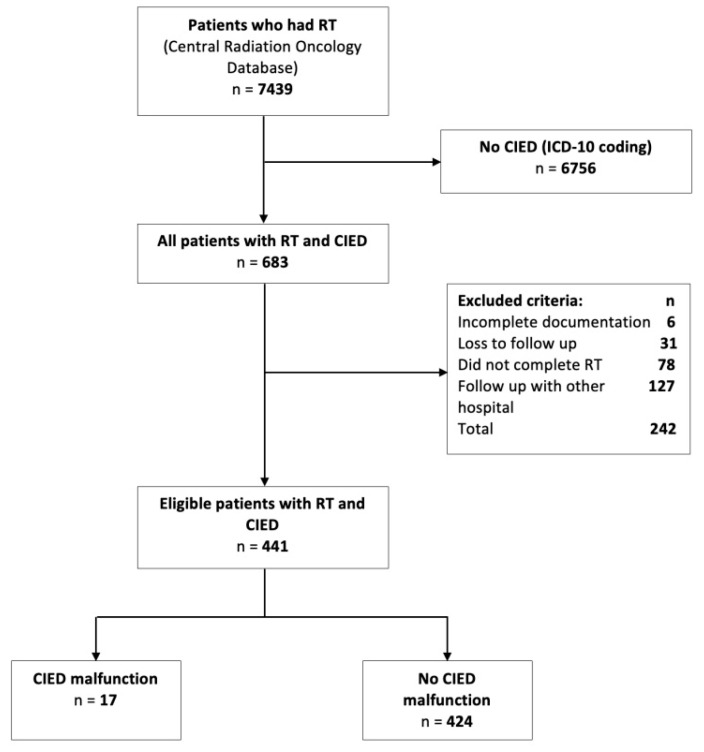
Study flow diagram showing patient selection, exclusions, and distribution of patients. CIED, cardiac implantable electronic device; ICD-10, The International Classification Diagnosis of Disease 10th Version; RT, radiotherapy. Central illustration: Incidence and predictors of cardiac implantable electronic device malfunction (CIED) with radiotherapy (RT). There were 17 patients with CIED malfunction out of 441 patients who underwent RT. Patients with CIED malfunction had a higher median RT cumulative dose (42.5 Gy vs. 36 Gy) and beam energy dose (14 vs. 6.75 MV). Types of CIED malfunction ranges from minor electrical reset, noise oversense with no symptoms, change in lead parameters and unexpected decline in battery life. Adjusted logistic regression analysis identified that CIED from ICD device class, anatomical RT field above the diaphragm and usage of photon beam energy ≥ 10 MV were predictors of CIED malfunction with RT. There was no significant correlation between cumulative RT dose and CIED malfunction. A and P, abdomen and pelvis; CIED, cardiac implantable electronic device; H and N, head and neck; PPM, permanent pacemaker; Gy, Gray; ICD, implantable cardioverter defibrillator; IQR, interquartile range; LL, lower limb; MV, megavolt; RT, radiotherapy; UL, upper limb.

**Table 1 jcm-11-06329-t001:** Comparison between patient characteristics, CIED data and RT details in patients with CIED malfunction vs. no CIED malfunction with RT. CIED, cardiac implantable electronic device; CRT–P, cardiac resynchronization therapy pacemaker; CRT–D; cardiac resynchronization therapy defibrillator; DC PPM, dual chamber permanent pacemaker; Gy, gray; ICD, implantable cardioverter defibrillator; MV, megavolt; RT, radiotherapy; SC PPM, single chamber permanent pacemaker.

Category	Characteristics	All *n* = 441	CIED Malfunction *n* = 17	No CIED Malfunction *n* = 424	*p* Value
Patient characteristics	Age, years (±SD)	82 ± 14	85 ± 11	82 ± 14	0.12
	Male gender, *n* (%)	327 (74)	14 (82)	313 (74)	0.061
Device type *n* (%)	SC PPM	41 (9)	0 (0)	41 (9)	<0.001
	DC PPM	303 (69)	5 (29)	298 (70)	0.024
	CRT–P	9 (2)	1 (6)	8 (2)	0.032
	ICD	44 (10)	8 (47)	36 (8)	0.047
	CRT–D	44(10)	3 (18)	41 (10)	0.014
	Age of device, *n* (IQR)	5.2 (0.9–11.2)	8 (4.7–11.2)	5.1 (0.9–10.9)	0.072
Device manufacturer *n* (%)	Medtronic	198 (45)	9 (54)	189 (45)	0.016
	St Jude	129 (29)	4 (21)	125 (29)	0.013
	Boston Scientific	92 (21)	3 (19)	89 (21)	0.021
	Biotronik	22 (5)	1 (6)	21 (5)	0.045
Safety measures *n* (%)	Device dependency	216 (49)	10 (59)	206 (49)	0.039
	Total safety measures	84 (19)	0 (0)	84 (20)	<0.001
	Application of magnet	49 (11)	0 (0)	49 (12)	<0.001
	Device reprogramming	35 (8)	0 (0)	35 (8)	<0.001
	Surgical relocation of CIED	0 (0)	0 (0)	0 (0)	N/A
RT anatomical location *n* (%)	Head and neck	137 (31)	3 (15)	134 (32)	0.021
	Thorax	88 (20)	12 (71)	76 (18)	0.014
	Abdomen and pelvis	132 (30)	2 (14)	129 (31)	0.035
	Spine	31 (7)	0 (0)	31 (7)	<0.01
	Upper limbs	22 (5)	0 (0)	22 (5)	<0.01
	Lower limbs	31 (7)	0 (0)	31 (7)	<0.01
RT details, *n* (IQR)	Cumulative dose, Gy	36 (1.8–80)	42.5 (20–77)	36 (1.8–80)	0.39
	Dose per fraction, Gy	3.75 (1–26)	12 (10–26)	3.6 (1–24)	0.26
	Number of fractions	13 (1–40)	6 (2–23)	12.9 (1–40)	0.42
	Dose to device, Gy	0.28 (0–3.3)	0.29 (0.1–3.3)	0.26 (0.1–3)	0.13
Beam energy, *n* (%)	Total beam energy used	423 (96)	16 (95)	406 (96)	0.037
	Photon ≥ 10 MV	185 (44)	16 (95)	169 (40)	0.045
	Photon < 10 MV	145 (34)	0 (0)	145 (34)	<0.01
	Electron	93 (22)	0 (0)	93 (22)	<0.01
	Photon dose in MV, *n* (IQR)	7 (5–18)	14 (12–18)	6.75 (5–15)	0.037
	Electron dose in MV, *n* (IQR)	9 (3–18)	0	9 (3–18)	<0.01

**Table 2 jcm-11-06329-t002:** Types of CIED malfunction detected in patients who underwent RT. PPM group consists of SC PPM, DC PPM, and CRT–P. ICD group consists of ICD and CRT–D. ATP, antitachycardia pacing; CIED, cardiac implantable electronic device; CRT–P, cardiac resynchronization therapy pacemaker; CRT–D; cardiac resynchronization therapy defibrillator; DC PPM, dual chamber permanent pacemaker; ICD, implantable cardioverter defibrillator; SC PPM, single chamber permanent pacemaker.

Types of CIED Malfunction	PPM Group, (%)	ICD Group, (%)	*n* (%)
Minor electrical reset	0 (0)	5 (45)	5 (29)
Major electrical reset	0 (0)	0 (0)	0 (0)
Unexpected decline in battery	3 (50)	1 (10)	4 (24)
Loss of telemetry	0 (0)	0 (0)	0 (0)
Change in lead parameters	2 (33)	0 (0)	2 (12)
Noise oversense with no symptoms	1 (17)	5 (45)	6 (35)
Noise oversense with inhibition of pacing and/or symptoms	0 (0)	0 (0)	0 (0)
ATP treatment	0 (0)	0 (0)	0 (0)
Inappropriate shock	0 (0)	0 (0)	0 (0)
Total CIED malfunction	6 (35)	11 (65)	17 (100)

**Table 3 jcm-11-06329-t003:** Crude and adjusted logistic regression analysis of factors associated with CIED malfunctions with RT (*n* = 441). PPM group consists of SC PPM, DC PPM, and CRT–P. The ICD group consists of ICD and CRT–D. The anatomical location of RT above the diaphragm consists of the head and neck, thorax, and upper limb. The anatomical location of RT below the diaphragm consists of the abdomen and pelvis, spine, and lower extremities. CIED, cardiac implantable electronic device; CRT–P, cardiac resynchronization therapy pacemaker; CRT–D; cardiac resynchronization therapy defibrillator; DC PPM, dual chamber permanent pacemaker; Gy, gray, ICD, implantable cardioverter defibrillator; MV, megavolt.

Variable	Crude OR (95% CI)	Adjusted OR (95% CI)	*p*-Value for Adjusted OR
Device class: ICD group vs. PPM group	4.6 (0.75–10.2)	4.8 (0.84–11.1)	0.021
Anatomical location of RT (above vs. below diaphragm)	5.2 (1.82–15.2)	4.8 (0.82–8.25)	0.014
Cumulative tumor dose (10 Gy increment)	1.20 (0.95–1.52)	1.13 (0.89–1.44)	0.31
Fraction dose (1 Gy increment)	1.0 (0.79–4.7)	1.1 (0.84–4.98)	0.43
Photon beam energy ≥10 MV vs. <10 MV	6.73 (1.58–10.76)	6.91(1.67–11.85)	0.007

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
