# Peer review of "Incidence and Predictors of Cardiac Implantable Electronic Devices Malfunction with Radiotherapy Treatment"

_jcm, 2022, doi:10.3390/jcm11216329_

Round 1
Reviewer 1 Report
The authors reported their experience of 441 patients with CIED who underwent RT. CIED encountered was permanent pacemaker (PPM) (78%), cardiac resynchronization therapy – pacing (CRT-P) (2%), implantable cardioverter defibrillator (ICD) (10%) and CRT-defibrillator (CRT-D) (10%).
They estimated the mean cumulative dose and the dose where 17 devices had malfunctioned. They found that Using logistic regression, the predictors of CIED malfunction were photon beam energy use 10 MV (OR 5.73; 95% CI, 1.58 – 10.76), anatomical location of RT above the diaphragm (OR 5.2, 95% CI, 1.82 – 15.2) and having a CIED from the ICD group (OR 4.6, 95% CI, 0.75 – 10.2)
They concluded that clinicians involved in taking care of these patients should be aware of the predictors of CIED malfunction with RT to ensure the safety of oncology patients. The interesting findings: cumulative RT dose has no significant correlation with device malfunction despite the traditional assumption that RT causes malfunctions CIED during treatment.
I congratulate the authors for this nice study.
Weakness:
1. Single-center retrospective study
2. Did not discuss the time of chemotherapy the patients had and the impact of chemotherapy on CIED if any?
4. No information on cardiac function
5. Any impact of renal dysfunction or electrolyte imbalance on arrhythmias and hence CIED?
Strengths:
1. Granular Data
2. Compared PPM with ICD (table-1)
3. Categorized CIED malfunctions during radiotherapy systematically (Summary-1)
4. Used appropriate statistics and defined the risk of RT dose
5. Discussion is appropriate with adequate citation and scientifically sound
6. Summarized the recommendations
7. Liked the clinical perspectives and future directions.
Author Response
Thank you for the positive and encouraging feedback on our study. The answers to the weakness of this study are:
- We were limited in our resources with this study. With the result of this published study, we hope to expand this study with a sub-study with more support after proving our credibility.
- The time of chemotherapy was not included as we believe that it did not add much to the clinical effect of cardiac dysfunction with radiotherapy. We are in the midst of collating duration/timing of chemotherapy resulting in types of cardiac dysfunction
- We did not include cardiac function as we believe that it won't affect the types of cardiac dysfunction. If there were any symptomatic clinical CIED dysfunction, we were planning to include cardiac function in our study to see if that affects clinically visible device dysfunction.
- As there were no apparent arrhythmias in our study ( just oversense), it would be difficult to assess if electrolyte disturbances caused arrythmias in this study
Reviewer 2 Report
The Authors investigate the incidence of cardiac implantable electronic devices (CIED) malfunction with radiotherapy (RT) treatment and assess predictors of CIED malfunction. In 6-year retrospective analysisidentified 441 patients with CIED who underwent RT. CIED encountered was permanent pacemaker (PPM) (78%), cardiac resynchronization therapy–pacing (CRT-P) (2%), implantable cardioverter defibrillator (ICD) (10%) and CRT-defibrillator (CRT-D) (10%).
The baseline characteristics, CIED data, and RT details of all patients with CIED who underwent RT were reported. The study demonstrates that photon beam energy >10 MV, RT above the diaphragm, and ICDs are significantly associated with CIED malfunction. Traditionally, cumulative RT dose was thought to be the predominant cause of CIED malfunction. The results highlight the need to observe beam energy dose and to institute re-planning of radiotherapy to reduce the risk of CIED malfunction.
The authors conducted this retrospective study with scientific rigor. All aspects concerning the topic in question were evaluated and final results were given with feedback in clinical practice. The same authors report the limitations of the study and the need for new studies that take into account their results. The studies are valid to lay the foundations for a new vision of the strongly ingesting problem in radiotherapy treatments. Thank you for allowing me to perform this interesting review.
Author Response
Dear reviewer,
Thank you for your excellent and encouraging feedback. We do hope that this study will assist in the management of this patient cohort